# Explainable Machine Learning with Pairwise Interactions for Predicting Conversion from Mild Cognitive Impairment to Alzheimer’s Disease Utilizing Multi-Modalities Data

**DOI:** 10.3390/brainsci13111535

**Published:** 2023-10-31

**Authors:** Jiaxin Cai, Weiwei Hu, Jiaojiao Ma, Aima Si, Shiyu Chen, Lingmin Gong, Yong Zhang, Hong Yan, Fangyao Chen

**Affiliations:** 1Department of Epidemiology and Biostatistics, School of Public Health, Xi’an Jiaotong University, Xi’an 710061, China; mathcjx@stu.xjtu.edu.cn (J.C.); xjhww2016@stu.xjtu.edu.cn (W.H.); siaima@stu.xjtu.edu.cn (A.S.); shiyu_chen@stu.xjtu.edu.cn (S.C.); gonglingminn@stu.xjtu.edu.cn (L.G.); 2Department of Neurology, Xi’an Gaoxin Hospital, Xi’an 710077, China; jma9211@126.com; 3Department of Surgical Oncology, First Affiliate Hospital of Xi’an Jiaotong University, Xi’an 710061, China; yongzhang761@xjtu.edu.cn; 4Key Laboratory for Disease Prevention and Control and Health Promotion of Shaanxi Province, Xi’an Jiaotong University, Xi’an 710061, China; 5Department of Radiology, First Affiliate Hospital of Xi’an Jiaotong University, Xi’an 710061, China

**Keywords:** interpretable machine learning, explainable boosting machine, multimodality, interaction, Alzheimer’s disease

## Abstract

Background: Predicting cognition decline in patients with mild cognitive impairment (MCI) is crucial for identifying high-risk individuals and implementing effective management. To improve predicting MCI-to-AD conversion, it is necessary to consider various factors using explainable machine learning (XAI) models which provide interpretability while maintaining predictive accuracy. This study used the Explainable Boosting Machine (EBM) model with multimodal features to predict the conversion of MCI to AD during different follow-up periods while providing interpretability. Methods: This retrospective case-control study is conducted with data obtained from the ADNI database, with records of 1042 MCI patients from 2006 to 2022 included. The exposures included in this study were MRI biomarkers, cognitive scores, demographics, and clinical features. The main outcome was AD conversion from aMCI during follow-up. The EBM model was utilized to predict aMCI converting to AD based on three feature combinations, obtaining interpretability while ensuring accuracy. Meanwhile, the interaction effect was considered in the model. The three feature combinations were compared in different follow-up periods with accuracy, sensitivity, specificity, and AUC-ROC. The global and local explanations are displayed by importance ranking and feature interpretability plots. Results: The five-years prediction accuracy reached 85% (AUC = 0.92) using both cognitive scores and MRI markers. Apart from accuracies, we obtained features’ importance in different follow-up periods. In early stage of AD, the MRI markers play a major role, while for middle-term, the cognitive scores are more important. Feature risk scoring plots demonstrated insightful nonlinear interactive associations between selected factors and outcome. In one-year prediction, lower right inferior temporal volume (<9000) is significantly associated with AD conversion. For two-year prediction, low left inferior temporal thickness (<2) is most critical. For three-year prediction, higher FAQ scores (>4) is the most important. During four-year prediction, APOE4 is the most critical. For five-year prediction, lower right entorhinal volume (<1000) is the most critical feature. Conclusions: The established glass-box model EBMs with multimodal features demonstrated a superior ability with detailed interpretability in predicting AD conversion from MCI. Multi features with significant importance were identified. Further study may be of significance to determine whether the established prediction tool would improve clinical management for AD patients.

## 1. Introduction

Alzheimer’s disease (AD) is a degenerative chronic brain disease that primarily affects individuals above 65 years old. According to the World Health Organization (WHO), approximately 50 million individuals are living with AD, and this number is expected to triple by 2050 [1]. Unfortunately, there is currently no cure for AD, and existing treatments can only help slow its progression. It is essential to diagnose AD at an early stage, as the available treatment options are most effective during the early stages of the disease [2].

Mild Cognitive Impairment (MCI) can be regarded as an early stage of AD or pre-AD. Over 33% of MCI patients will progress to AD within five or more years [3]. Thus, predicting the progression from MCI to AD is crucial for effective treatment and would benefit the well-being of AD patients, as well as their families [4].

Magnetic Resonance Imaging (MRI)-based markers have gained attention in recent decades for the diagnosis of AD and predicting the conversion from MCI to AD, which is a typical multi-modal data in clinical practice [5].

The use of multi-modal data for building diagnostic systems has been highly encouraged because it enhances predictive performance [6]. In order to processing the multi-modal data, numerous machine learning (ML) techniques, especially deep learning techniques, have been used for identifying the progress of AD and predicting the converting to AD from MCI [7]. However, the practical application of ML-based prediction systems in the medical scenario has been hindered by its neglect of interpretability concerns, as complex models usually tend to sacrifice interpretability for accuracy.

Clinical experts are hesitant to trust black-box models that lack comprehensive and easy-to-understand explanations, despite their high performance [8]. Therefore, balancing interpretability and accuracy is crucial in various fields, especially in the medical field [9]. Recent advancements in eXplainable Artificial Intelligence (XAI) provide methods for understanding complex models and explaining their decisions, so as to bridge the gap between academic research and effective utilization in medical practice [10].

The Explainable Boosting Machine (EBM) model is one of them. The interpretability provided by the EBM model comes from its own mathematical formula and does not require the use of other values, making it inherently explainable. Moreover, it can ensure performance metrics comparable to complex black-box models. Furthermore, EBM can also take into account the interaction effects of certain factors [11].

Some researchers have applied the EBM model to predict severe retinopathy of prematurity or Parkinson’s and other diseases, achieving model interpretability while ensuring accuracy [12,13]. However, with the literature review, we found there is only limited research on predicting the conversion from MCI to AD using the EBM model. Moreover, they only had a relatively short follow-up time or just one follow-up visit, making it hard to achieve long-term prediction [14].

To improve the accuracy of predicting MCI to AD conversion, it is necessary to consider various factors that may impact the prediction model. Future research could investigate the potential interaction effects of demographic factors on prediction accuracy and explore additional relevant factors to enhance the model’s performance.

In this study, we aimed to establish an EBM model to predict whether patients with MCI will convert to AD in future follow-up periods (i.e., at 1, 2, 3, 4, and 5 years) using the data obtained from the Alzheimer’s Disease Neuroimaging Initiative (ADNI) database. 

The main contributions of this article are as follows: (I) We evaluate, for the first time, the performance of EBM predicting the conversion from MCI to AD during a follow-up period of 1–5 years by using MRI, cognitive measures, and social–demographical–clinical measurements versus using only one of the two modalities. (II) We investigate the changing of importance of each feature at different follow-up stages. (III) We provide the visualized results about the contributions of each factor to the conversion from MCI to AD, as well as possible interactive contributions. (IV) We offer local explanations for each individual’s prediction decision.

## 2. Method

### 2.1. Data Source

The data involved in this study were obtained from the Alzheimer’s Disease Neuroimaging Initiative (ADNI) database (https://adni.loni.usc.edu (accessed on 22 April 2022)). The ADNI database was launched as a public–private partnership in 2003. The primary objective of ADNI database has been to assess whether a combination of MRI, PET, clinical, neuropsychological assessments, and other biological markers can effectively measure the progression of aMCI and early AD. The ADNI project was approved by the Review Board of each participant site, and all participants provided written informed consent at the time of enrollment, including permission for data sharing and analysis [15].

The samples involved in this study was consisted of 1042 Amnestic MCI individuals at baseline from ADNI1, ADNI2, ADNI3, and ADNIGO, which are different phases of ADNI program. Our feature set includes baseline demographic data such as age, gender, years of education, as well as clinical such as APOE4 status, ADAS13, MMSE, CDRSB, and MRI-related measures, which were used as inputs to the EBMs [16,17,18,19]. The response variable is the cognitive status diagnosis of each patient during a specific follow-up period in the future. As for the inclusion and exclusion criteria for the participation, we only chose those aMCI participations with complete data

### 2.2. MRI Image Preprocessing

The MRI images were processed with FreeSurfer software (version 7.3.2) using the standard cross-sectional pipeline. The preprocessing of the MRI image was achieved through nonparametric nonuniform intensity normalization (N3)-based bias field correction. To ensure consistency across all images, registration was performed to ensure that they were in the same orientation and roughly the same spatial correspondence. After brain extraction and affine transformation, all images were reviewed by a well-trained professional who visually inspected them. Scans that had severe MRI artifacts, brain extraction failure, or poor registration were excluded from further analysis. By following these preprocessing steps, we ensured that the MRI images used in our study were of high quality and free from potential sources of bias [20].

### 2.3. Statistical Analysis

#### 2.3.1. Feature Selection

According to our research purpose and the literature review, relevant factors taken into consideration can be classified into three modalities including scores of multi-types of neurocognitive scales, MRI measurements, and social–demographical–clinical features [16,17,18,19].

For scores of neurocognitive scales, we explored the potential prognostic value of baseline neurocognitive scores obtained from the ADNI dataset. Specifically, we included scores from the ADAS-cog-13, and the Mini-Mental State Examination (MMSE), Functional Activities Questionnaire (FAQ), as well as the Clinical Dementia Rating Scale Sum of Boxes (CDR-SB). For MRI measurements, we selected the MRI results that may be related to AD. They contain the volume of left and right hippocampus and amygdala, and the volume and thickness of left and right entorhinal and inferior temporal. Social–demographical–clinical features contained the individuals’ age, gender, education attainment, diagnose at baseline (early or late MCI, abbreviated as DX_bl), and APOE4. In total, we included 21 relevant factors. 

Previous research has also suggested that the prognostic value of neurocognitive scores may vary, depending on the remaining time to the onset of dementia [21]. Thus, the outcomes in this study are defined as the future clinical statuses (convert to AD or not) during follow-up within 1, 2, 3, 4, and 5 years of the same population. It is worth mentioning that other dementia types were excluded from the analyses. This is to explore the relationship between influential variables and response variables in our study to gain a better understanding of the potential utility of these measures in predicting future cognitive decline.

#### 2.3.2. Explainable Machine Learning Analysis

This study employed interpretable ML methods to analyze the factors influencing the conversion from MCI to AD over different follow-up periods and predict the likelihood of such conversion within different follow-up intervals. The EBM algorithm was utilized in this study. The EBM is an explainable ML algorithm that combines Generalized Additive Models (GAMs) with gradient boosting [11].

GAMs are types of regression models that allow for flexible modeling of nonlinear relationships between the response variable and the relevant variables [22]. They model the relationship between the response variable *y* and the relevant variables *x* as a sum of smooth functions *s*. The smooth functions *s* is typically modeled using splines or other smoothing functions. The *j* indexes the relevant variables, *g* is the link function that adapts the GAMs to different settings such as regression or classification as in Equation (1): (1)$$g(E[y])=\beta_{0}+\sum s_{j}\left(x_{j}\right)$$

However, GAMs are limited to modeling only main effects, and do not account for interactions between variables.

The EBM algorithm extends GAMs by adding pairwise interactions between relevant variables, taking the name of GA^2^M, which can be defined as in Equation (2), where the $s_{i_{j}}\left(x_{i},x_{j}\right)$ donates the pairwise interactions [23].
(2)$$g(E[y])=\beta_{0}+\sum s_{j}\left(x_{j}\right)+\sum s_{i_{j}}\left(x_{i},x_{j}\right)$$

The two-dimensional term $s_{i_{j}}\left(x_{i},x_{j}\right)$ can relate the response variable to pairs of independent variables. These interactions are modeled using decision trees, which are then combined with GAMs using gradient boosting algorithm. 

The gradient boosting is a machine learning technique that sequentially adds weak learners (i.e., decision trees with shallow depth, or linear models) to the model, with each learner focusing on the errors made by the previous learners [24]. The resulting model is a boosted ensemble of decision trees and GAMs, which can capture complex nonlinear relationships and interactions between variables [25].

Moreover, EBMs are highly intelligible. The model produces transparent models that can be easily understood by researchers. The EBM algorithm provides global feature contribution that can be used to identify the most important relevant factors influential factors in the model. By exploiting the additivity and modularity of these contributions, it becomes possible to rank and visualize which features have the highest impact on the model’s prediction [26].

EBMs not only provide a global interpretation of their predictions, but also offer local interpretations by quantifying the contribution of each feature to the final prediction of each subject [27]. To evaluate the local explanation of test participants, the most important features in a single prediction were ranked. This ranking was obtained by calculating the logit of the probability, which corresponds to the logarithm of the odds, using the logistic link function *g* (Equation (2)). The final prediction of EBMs was obtained by summing the logit of each feature. 

Such a method enables medical experts to identify which features increase or decrease the predicted probabilities made by the model. The EBMs strives to offer a fully interpretable learning framework, as different to the technique of enhancing interpretability for a black box classifier, such as SHAP or LIME. This approach can facilitate the comprehension of the factors influencing the predicted risk of a particular outcome for patients, thereby enabling healthcare providers to enhance their decision-making processes [28].

Figure 1 illustrates the flowchart of this study. The dataset is divided into training and test sets with a percentage, respectively, of 90% and 10%. In order to handle the issues of data imbalance, the synthetic minority oversampling technique (SMOTE) is used. In each follow-up period, we trained three EBM classifiers based on the following three feature combinations: 

Comb. (a) both MRI-driven biomarkers and cognitive test scores, plus age, gender, education, diagnose at baseline, and APOE4. 

Comb. (b) Cognitive test scores, plus age, gender, education, diagnose at baseline, and APOE4. 

Comb. (c) MRI-driven biomarkers, plus age, gender, education, diagnose at baseline, and APOE4. 

The baseline information was used as input of all EBMs and the converting AD or not during follow-up period was used as output. All EBMs were tested with the 5-fold cross validation. 

The classifications performances were measured by accuracy, specificity, sensitivity, and area under the receiver operating curve (AUC). Then, we used the trained EBM classifier with feature sets in Comb. (a) in test sets to achieve the global explanation and local explanation in each follow-up period. 

All analyses were conducted with Python 3.9 and the package InterpretML 0.3.0, which implements the EBM algorithm, on Windows 11 (2.10 GHz, 16 GB of RAM).

## 3. Results

### 3.1. Characteristics of Included Patients

At baseline, a total of 1042 patients with MCI were included in the study. The majority of patients were male (58%) and the median age was 73 years (range: 68–78 years). The median education level was 16 years (range: 14–18 years). The age at baseline, gender, education at baseline, MMSE, CDRSB, and ADAS13 was almost homogenous. The dementia individual percentage was much larger in long-term follow-up period. Table 1 reports the demographic and the clinical information of the participants’ included in this study. 

We also investigate the corresponding information at follow-up period within 1, 2, 3, 4, and 5 years. At each visit of the ADNI study, patients were evaluated for AD based on NINCDS-ADRDA criteria [29]. Other dementia types were not taken into account. The AD percentages were 12%, 27%, 32%, 36%, and 38% within 1, 2, 3, 4, and 5 years follow up periods, respectively, with an increasing trend.

### 3.2. Classification Performances

The model accuracy, sensitivity, specificity, and AUC value at each follow-up time point are plotted in Figure 1 for the three feature combinations: Comb. (a), Comb. (b), and Comb. (c)

As shown in Figure 2, the classification AUC value derived from the joint use of Comb (a) were found to be significantly higher than those of other feature combinations. The mean AUC using Comb (a) displayed a characteristic pattern of first at about 0.9, then decreasing, reaching a low point at 3 years, followed by a gradual increase until 5 years of follow-up, to about 0.92. 

A similar trend was observed in the lines of the Comb (c), although with different magnitudes, and the line of Comb (a) being slightly smaller than that of Comb (c).

During the short term of follow-up (e.g., one-year follow up), the models utilizing Comb (a) and Comb (c) exhibited superior performance across nearly all metrics compared to the model utilizing Comb (b). However, during the mid-term follow-up period, such as 2 or 3 years, nearly all performances of the three classifiers were lower than those of other follow-up periods and showed a certain degree of false positives.

In the long-term follow-up period, the specificity of Comb (b) exceeded that of using Comb (a) and Comb (c), while the sensitivity of using Comb (c) exceeded that of the others. Nonetheless, the AUC is a more informative metric for evaluating imbalanced data classification, as it takes into account both true positive rate (sensitivity) and false positive rate (1-specificity) across different probability thresholds. 

Overall, a high AUC value indicates a good balance between sensitivity and specificity, regardless of class distribution imbalance. According to the AUC value, the Comb (a) outperformed the other feature combinations’ performances in all periods (Figure 2d).

### 3.3. Global and Local Explanation Learning Analysis

By utilizing EBMs model, we can not only obtain more accurate predictions, but also find out why the model gives such prediction results, which variables play the main role, and what proportion. Because the Comb (a) shows a relative better performance, we chose this feature set to exhibit the following explainable learning analysis results.

The explanation contains global and local explanations, which are two critical concepts in the field of explainable machine learning. It typically includes information about important features, their relationships, and how they influence the model’s predictions. In contrast, local explanation is instance-specific and provides insight into why a particular prediction was made for a specific input. It highlights the most important features that influenced the model’s decision and explains how they contributed to the prediction [30].

#### 3.3.1. Global Explanation

##### Feature Importance

Global explanation provides an overall understanding of how a machine learning model operates, offering a high-level summary of the model’s behavior. The importance of each feature in predicting MCI individuals converting or not at each follow-up is shown in Figure 3. In terms of MRI imaging data and cognitive test data, MRI imaging occupies a more prominent position in both the short and long term follow up periods, with a significantly higher proportion of top three rankings than cognitive tests, while in the mid-term follow-up period, cognitive scale score is more important, with a higher proportion of top three rankings compared to MRI imaging.

We also found that the importance rankings of the volume of the inferior temporal and entorhinal are higher than that of their thickness in most cases.

For the short-term follow-up period, the inferior temporal was more important in the imaging indicators. For the long-term follow-up period, the entorhinal cortex was more important.

##### Uni-Factor Interpretation

Figure 4 illustrates the feature interpretability plots for the most significant variables in predicting the follow-up periods. These plots, also known as risk profiles, depict the risk score on the vertical axis and the actual value of the feature on the horizontal axis (upper graphs in Figure 4). The bottom graphs in Figure 4 display the density or distribution of the feature. A feature risk score above zero indicates a contribution to the positive class classification (i.e., converting to AD), whereas a score below zero indicates a contribution to the negative class classification (i.e., not converting to AD).

In the one-year follow-up period, lower right inferior temporal volume values (<9000) are the most significant relevant factors of AD conversion (Figure 4a). During the two-year follow-up period, a left inferior temporal thickness value lower than 2 is the most critical feature in predicting AD conversion (Figure 4b). For the three-year follow-up period, higher FAQ scores (>4) emerge as the most important factor in predicting AD conversion (Figure 4c). In four years of follow-up, the APOE4 count of 1 or higher is the most significant relevant factor of AD conversion (Figure 4d). Finally, in five years of follow-up, lower right entorhinal volume values (<1000) are the most critical feature in predicting AD conversion (Figure 4e).

##### Analysis of Interaction-Effects 

As shown in Figure 3c, we noticed there are many pairwise interactions in the three-year period prediction, and the AGE and APOE4 interaction is in the top 5, followed by the DX_bl (Diagnose at baseline) and CDRSB interaction in the top 6, as shown in Figure 5, which is the heat map of the two pair interaction. 

The closer the color is to yellow (indicating a positive score), the higher the risk of conversion to AD. Conversely, the closer the color is to blue (indicating a negative score), the lower the likelihood of conversion to AD. The heat map of AGE and APOE4 interaction indicates that having 2 APOE4 and aging over 75 years old results to having higher risk to convert to AD in three-year period, as shown in Figure 5a.

The heat map of DX_bl and CDRSB interaction (Figure 5b) indicates being diagnosed as LMCI at baseline, and a CDRSB score greater than 5 results in having a higher risk to convert AD in a three-year period. These two parts can be clearly seen in the figure close to yellow (a positive score).

#### 3.3.2. Local Explanation

The local explanation focuses on training the local surrogate model to interpret the individual predictions. According to the local explanation results, we can figure out what role each variable plays in the prediction (positive or negative) and its magnitude. Due to the large number of patients, we cannot introduce them one by one. Thus, we selected four patients in the three-year follow up period prediction ramdomly, two of which were diagnosed with AD and two who were not diagnosed with AD, and the four were all predicted correctly. In these figures, the bar of each variable facing to the right indicates support for a prediction of 1: that this patient will convert to AD within three years. The bar facing to the left is opposite. We can find that most of the variables support this patient to convert to AD within three years in terms of Patients 1 and 2, and the CDRSB and FAQ played an important role in their correct predictions (Appendix A). As for Patient 3 and 4, most variables support this patient will not convert to AD within three years, and the interaction of DX_bl and CDRSB played a significant role in their correct predictions (Appendix A).

## 4. Discussion

The early detection of AD is clinically valuable to stop its progress at the early stages and improve patients’ and their relatives’ quality of life. Our work is primarily based on explainable EBM models that utilize multimodal features as inputs to predict whether patients with MCI will convert to AD during follow-up periods of varying lengths. In this study, we compared different modalities’ combinations, and found that in terms of accuracy, sensitivity, specificity, and AUC, the use of Comb a) demonstrated superior performance, for the most part, overusing a single modality. Especially in AUC and accuracy, the superiority of utilizing Comb a) was consistently observed throughout the entire follow-up period (Figure 2). 

As shown in Figure 2, it is noteworthy that when predicting the progression of MCI in a three-year period, the performance of the model is relatively less satisfactory compared to other follow-up periods, particularly with some false positives. We assume that the occurrence of this situation is due to a relatively brief follow-up period that did not afford enough time for all prodromal AD participants to progress to a clinical diagnosis of dementia: these false positives represent individuals whose diagnostic classification did not change during the short-term follow-up period, despite the disease progressing and eventually reaching the dementia stage at a later time point. To validate this hypothesis, we examined the disease progression in these false positive cases. The results showed that 50% of false-positive MCI patients would convert to AD at the 5-year follow-up, which is almost four times the conversion rate to AD in the MCI population [31]. To be more specific, when our model makes a false prediction of conversion to dementia within 3 years, it is likely indicative of pathophysiological progression in the brain, but it may require more time for the disease to advance to the dementia stage.

Some may question that, according to this hypothesis, the prediction should have poor performance when forecasting for one-year follow up. It is well acknowledged that AD is a slowly progressive disease [32], and predicting at the first year is approximate to conducting a classification at the present time to determine whether a patient is MCI or AD. While to our knowledge, in recent years, there are many studies use different machine learning approach to address this problem, and achieve high quality classifications. Thus, to some extent, the feasibility of predicting AD or MCI in the one-year follow up period is achievable through technological means. 

Moreover, to our current knowledge, using only cognitive tests to classify MCI and AD at the present time cannot achieve a satisfactory level of accuracy. These also indirectly indicate that the ranking of MRI results is more important in our short-term predictions (e.g., one-year follow up) (Figure 3a). Actually, in our study, we found using only MRI for short term predictions (e.g., one-year follow up) also yielded satisfactory performance.

During a five-years follow-up period prediction, if we fix the occurrence of AD as a time point and trace back from it, this is equivalent to making predictions in the early stages of AD. As shown in Figure 3e, we discovered that the volume and thickness of entorhinal cortex rank high position. It is to say, the entorhinal cortex plays a crucial role in early AD prediction, which supports previous findings in pathology [33].

For example, the thinning of the entorhinal cortex is a structural biomarker that is sensitive to changes in AD over short periods of time and is closely related to the severity of AD [34]. Moreover, existing research has demonstrated that the structure and pathological damage of the entorhinal cortex play a significant role in the early memory impairment observed in AD. Structural imaging studies have revealed that entorhinal cortex atrophy occurs in the early stages of AD, with severe neuronal loss in the second and third cortical layers of the entorhinal cortex reaching 70% and 40% of the total number of neurons, respectively [35]. Furthermore, studies have shown that regional cerebral blood flow in the entorhinal cortex brain region is significantly reduced in the preclinical stages of AD [36]. These findings suggest that the entorhinal cortex exhibits structural and metabolic impairments in the preclinical stages of AD, earlier than other brain regions. Our study provides evidence from the perspective of machine learning for the important role of entorhinal in predicting Alzheimer’s disease. It suggests that the entorhinal region is more important in predicting the conversion status in the early stage of AD, as known from the prediction within a five-year follow up period in our study. Meanwhile, we also discovered the inferior temporal region is more important in predicting the conversion status of AD in the same period. The inferior temporal gyrus plays an important role in verbal fluency, a cognitive function affected early in the onset of AD [37]. Our findings also provide a machine learning explanation about it. 

However, although only using a cognitive test cannot complete the classification task satisfactorily, it is not to say cognitive test is meaningless for predicting MCI converting to AD. As shown in Figure 3c, at three years’ follow-up, the importance of the cognitive test gains the upper hand. 

Belleville et al. (2017) found measures of verbal memory and language tests have a high predictive value for the progression from mild cognitive impairment (MCI) to dementia [21]. To put it differently, basic screening instruments, such as the MMSE, exhibit adequate precision in forecasting transitions. Devanand et al. (2007) pointed out that the integration of these MRI volumes with age and cognitive measures results in remarkably high levels of predictive accuracy, which could potentially have significant clinical implications [17]. While this finding did not consider the time cumulative effect, in our study, by considering prediction at different follow-up periods, we revealed that incorporating demographic and cognitive measurements into MRI results could slightly improve the prediction performance for future one-year prediction. The improvement was more significant for other future time periods.

Based on the trade-off among importance feature ranking, performances, and the cost, we recommend, for MCI patients, using MRI for predicting dementia status as a relatively accurate and cost-efficient method of short-term and long-term prediction, and using cognitive measures for mid-term prediction.

In terms of a three-year follow-up period, we found many interactions in feature importance ranking (Figure 3c). Thus, we conducted a comparative study considering interaction effects versus not considering interaction effects in the test dataset. The main performance indicators, accuracy, sensitivity, specificity, and AUC, were 0.75, 0.78, 0.73, and 0.81 (with interaction effects) compared to 0.72, 0.76, 0.68, and 0.78 (without interaction effects), respectively. It can be found that considering interaction improves the prediction performance to a certain extent. Our consideration of interaction effects and their visual interpretation offers opportunities for etiological research, improving model interpretability by identifying complex relationships between independent variables. Moreover, incorporating interaction effects improves model performance.

The strength of this article is, although various research studies have been conducted to examine Alzheimer’s disease, their primary focus is on the accuracy of benchmark ML algorithms. Moreover, we also focus on the interpretability. 

Our study has demonstrated the interpretability of EBMs, incorporating both global and local interpretability techniques. Global interpretability offers a holistic understanding of the model’s behavior, while local interpretability explains results at an individual level. The EBMs results provide interpretable variable importance and interaction effects, aiding clinical decision making and alleviating concerns about the application of machine learning in healthcare.

To put it more broadly, utilizing the EBM model will increase the trust in Deep Learning. The more the public’s confidence in Deep Learning is, the more medical professionals will use it, allowing them to encourage innovation and accelerate the adoption of next-generation capabilities.

There are some limitations in the current study. In terms of variable selection, we will explain in the literature the variables that are important for predicting AD and include them in our study. In fact, we also attempted a data-driven approach to automatically selecting variables, but the results were not as good as using the recommended variables from the previous literature combined with manual screening. Our research is not only aimed at achieving good accuracy, but also concerns interpretability. We use the interpretable EBM method to rank the importance of the variables previously found in the literature, which can be seen as a refinement and extension of previous research. Furthermore, we plan to apply interpretable learning methods to investigate the impact of hippocampal subfield segmentation on MCI to AD prediction.

## 5. Conclusions

In conclusion, we utilized EBMs model to predict the conversion from MCI to AD at different follow-up periods and provided both global and local explanations. Our results showed that the best prediction performance was achieved by combining MRI measurements, cognitive tests, demographic, and clinical indicators. It may be helpful for early treatment interventions in order to slow cognitive decline and delay the onset of dementia.

Furthermore, regarding the importance of feature-ranking the performances we obtained, we advise clinicians to use different indicators for the prediction of cognitive impairment in different stages to maximize the benefits. 

## Figures and Tables

**Figure 1 brainsci-13-01535-f001:**
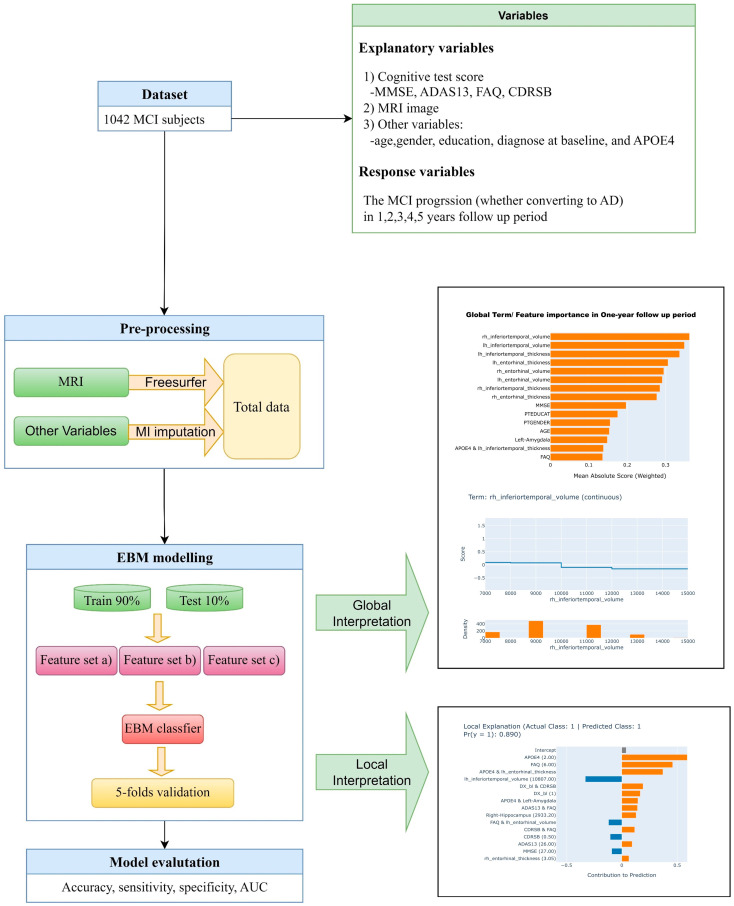
The flow chart of the whole study.

**Figure 2 brainsci-13-01535-f002:**
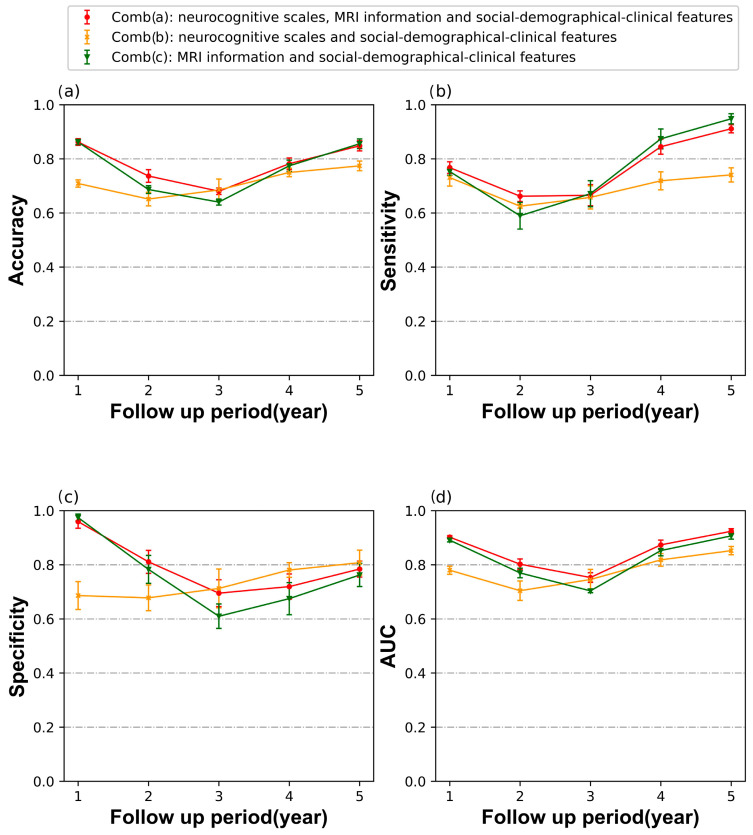
Accuracy, sensitivity, specificity, and AUC, with (**a**–**d**), for classifier performance from 1 to 5 years of follow-up. Error bars show the associated standard deviation.

**Figure 3 brainsci-13-01535-f003:**
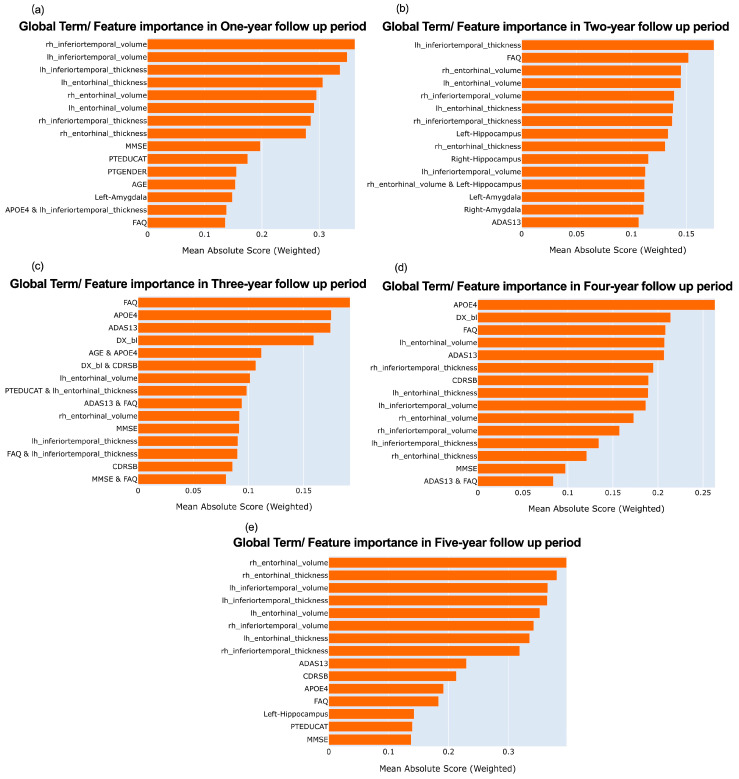
The rankings of the overall feature importance and mean absolute score from 1 to 5 years of follow-up. With (**a**–**e**) corresponding to 1, 2, 3, 4, and 5-year follow-up periods.

**Figure 4 brainsci-13-01535-f004:**
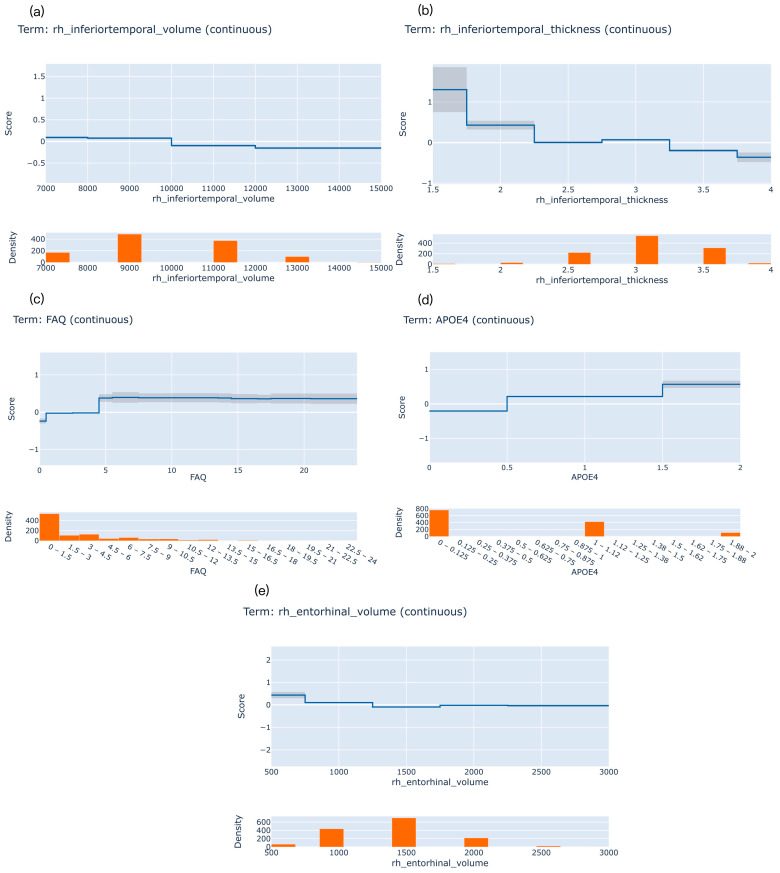
The uni-factor interpretation from 1 to 5 years of follow-up, with (**a**–**e**) corresponding to 1, 2, 3, 4, and 5-year follow up periods.

**Figure 5 brainsci-13-01535-f005:**
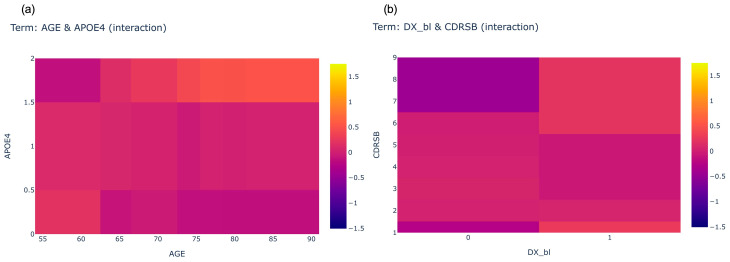
The heatmap of interaction ranking 5 (**a**) and 6 (**b**) in 3-year follow up period.

**Table 1 brainsci-13-01535-t001:** Participants information in different follow-up duration.

	Baseline	1 Year	2 Years	3 Years	4 Years	5 Years
DX						
- MCI	1042 (total)	778 (88%)	512 (73%)	392 (68%)	246 (64%)	140 (62%)
- Dementia		109 (12%)	191 (27%)	188 (32%)	140 (36%)	85 (38%)
AGE at baseline	73 (68,78)	74 (68,79)	73 (68,78)	73 (68,78)	73 (68,78)	73 (68,78)
GENDER						
- Female	428 (42%)	353 (40%)	290 (41%)	227 (39%)	150 (39%)	86 (38%)
- Male	614 (58%)	534 (60%)	413 (59%)	353 (61%)	236 (61%)	139 (62%)
Education at baseline	16 (14,18)	16 (14,18)	16 (14,18)	16 (14,18)	16 (14,18)	16 (14,18)
Median MMSE (First, third quantile)	28 (26,29)	28 (26,29)	27 (24,29)	27 (24,29)	27 (23,29)	26 (22,29)
Median FAQ (First, third quantile)	1 (0,5)	2 (0,7)	3 (0,10)	4 (1,13)	5 (1,15)	5 (1,18)
Median CDRSB (First, third quantile)	2 (1,2)	2 (1,2)	2 (1,4)	2 (1,4)	2 (1,5)	2 (1,6)
Median ADAS13 (First, third quantile)	16 (11,21)	17 (11,23)	18 (12,25)	18 (12,27)	18 (11,28)	18 (11,32)

Note: The DX and GENDER are reported in frequency (%), others are reported in median (First, third quantile).

## Data Availability

The de-identified data used in preparation of this article were obtained from the Alzheimer’s Disease Neuroimaging Initiative (ADNI) (https://adni.loni.usc.edu, Accessed: 22 April 2022). Details about data access are detailed there. The authors had no special access privileges others would not have to the data obtained from the Alzheimer’s Disease Neuroimaging Initiative (ADNI) database.

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
