# Peer review of "Explainable Machine Learning with Pairwise Interactions for Predicting Conversion from Mild Cognitive Impairment to Alzheimer’s Disease Utilizing Multi-Modalities Data"

_brainsci, 2023, doi:10.3390/brainsci13111535_

Round 1

Reviewer 1 Report

Comments and Suggestions for Authors

The authors apply the EBM model to predict the AD conversion from ACI based on feature combinations, and compare the performance and interpretability of the model in different follow-up periods. The features are a combination of MRI biomarkers, cognitive scores, social and clinical features. Through this, the authors argue that EBM offers satisfactory predictability and a detailed interpretability, and might enhance the adoption of machine learning in medicine. However, a few points if addressed would strengthen the work:

1. Ensure all acronyms are fully expanded upon their initial usage.

2. In the introduction, elaborate extensively on the rationale underpinning the adoption of Evidence-Based Medicine (EBM). Furthermore, present a concise yet detailed outline of the implementation strategy within the current scenario.

3. Within the "Data Source" section of the methods, provide an expanded explanation for the selection of classifiers, with a specific emphasis on the relevant sections of MRI images corresponding to distinct brain regions. Incorporate references that played a pivotal role in shaping these selections.

4. Address the lack of clarity in differentiating the superiority of one combination over the other within the figure. Deliver a succinct conclusion encapsulating the implications of the plot. If the models exhibit marginal divergence over the follow-up years, acknowledge this aspect.

5. Summarize the insights derived from the local explanations featured in the supplementary figure within a brief paragraph.

6. Dedicate a distinct section to elaborate upon the key findings presented in Figure 3.

7. Allocate a compact paragraph to elucidate how EBM can enhance adaptability within the medical community, utilizing your work as an exemplar.

Reviewer 2 Report

Comments and Suggestions for Authors

This study is a retrospective case-control study that employing the explainable machine learning models in predicting the conversion from MCI to Alzheimer's disease (AD). This study provide insight on the use of AI in disease prediction, which could be beneficial in the prevention and management of AD as AD has been known to be preventable in early stage. There are few comments from me to help improve the writing.

1. Please provide the full name for EBM in the abstract.

2. The first sentence of paragraph number 6 (line 81) in the introduction part is incomplete. Please double check.

3. In the methodology part under the subheading "Data sources", please explain what is "ADNI1, ADNI2, ADNI3, ADNIGO"? is this different database under ADNI or different study site?

4. Please state the inclusion and exclusion criteria for the participation in the ADNI (age, cognitive score and etc.).

5. Please provide a footnote in table 1 to indicate which data is presenting in median (range) or frequency (%). 

6. It would be great if the authors can increase the size of the wording to improve the readability of Figure 3. 

7. it would be great if the author could cite the more recent articles. Some of the cited articles were published 15 to 20 years ago. 

Overall, this is a good study. I would suggest the authors to cross validate the accuracy of this EBM model in the prediction of MCI conversion to AD using another study cohort or database in the future.

Reviewer 3 Report

Comments and Suggestions for Authors

This is a very interesting study employing an explainable ML algorithm to examine the predictive value of neurocognitive, demographic, and structural MRI variables for conversion to dementia in a large sample of patients with MCI. It is crucial, however, to review the state of art (previous studies using the ADNI or similar cohorts examining predictors of conversion to dementia using conventional statistical tools or ML). In general it is preferable to talk about probable AD, and aMCI throughout and be careful to distinguish between probable AD and dementia. 

Additional major comments

Global plots (figure 4) are valuable to aid interpretation of the role of specific predictor variables and help identify potentially useful cutoff values associated with increased likelihood of dementia conversion. It would help to compare variables though if subplots had the same vertical scale and the authors attempted a more clinical interpretation (e.g., RH IT volumes greater than <x value> were associated with a <Y%> increased risk of conversion at 1 year. How comparable are these estimated "cutoffs" across follow-up years for the same predictor variable? If they are not very comparable then some attempt to pool data from models a-e should be made (despite the obvious obstacle of including largely different subsamples in each set of models). 

On a related issue, to be able to compare predictor importance across follow-up periods, the subsamples used in models a-e should be comparable on key demographic (age, gender, education). Were they? Also how was (probable) AD diagnosed at each time point? were other dementia types excluded from the analyses?

Local explainability (using ceteris paribus plots) would be very interesting in addition to the single breakdown plot presented in the Supplementary Material with more extensive description to help the reader interpret the findings. I assume figure D.1 is from a participant who did not develop dementia at a specific follow-up? Additional examples (of a patient in Class 1 at the same time point) are needed. 

Overall, reducing the number of predictors to only 21 variables (out of the hundreds possible from the MRI data), limits the scope of the present work, although it makes the models simpler by not requiring the additional step of feature selection. Do the authors expect that model performance would be improved if additional features were included? 

Comments on the Quality of English Language

Several grammatical errors and oversights need to be corrected (e.g., lines 67, 69, 81, 137, 244, 246, 250, 251,  267, ..)
